# Co-Creating a Virtual Alcohol Prevention Simulation with Young People

**DOI:** 10.3390/ijerph17031097

**Published:** 2020-02-09

**Authors:** Lotte Vallentin-Holbech, Julie Dalgaard Guldager, Timo Dietrich, Sharyn Rundle-Thiele, Gunver Majgaard, Patricia Lyk, Christiane Stock

**Affiliations:** 1Centre for Alcohol and Drug Research, School of Business and Social Sciences, Aarhus University, 8000 Aarhus, Denmark; lvh.crf@psy.au.dk; 2Unit for Health Promotion Research, Department of Public Health, University of Southern Denmark, 6500 Esbjerg, Denmark; jguldager@health.sdu.dk; 3Research Department, University College South Denmark, 6100 Haderslev, Denmark; 4Social Marketing @ Griffith, Griffith Business School, Griffith University, Nathan QLD 4111, Australia and Centre for Youth Substance Abuse Research, Faculty of Health and Behavioural Science, University of Queensland, Brisbane QLD 4072; t.dietrich@griffith.edu.au (T.D.); s.rundle-thiele@griffith.edu.au (S.R.-T.); 5Embodied Systems for Robotics and Learning, The Maersk Mc-Kinney Moller Institute, University of Southern Denmark,5230 Odense, Denmark; gum@mmmi.sdu.dk (G.M.); pabl@mmmi.sdu.dk (P.L.); 6Charité—Universitätsmedizin Berlin, corporate member of Freie Universität Berlin, Humboldt-Universität Berlin, and Berlin Institute of Health, Institute for Health and Nursing Science, 13353 Berlin, Germany

**Keywords:** living lab methodology, co-creation, participatory research, empowerment, self-efficacy, alcohol prevention

## Abstract

Collaborative knowledge generation and involvement of users is known to improve health promotion intervention development, but research about the roles and perspectives of users in the co-creation process is sparse. This research aimed to study how young people perceived their involvement in a co-creation process focussed on the development of a gamified virtual reality (VR) simulation—VR FestLab. The Living Lab methodology was applied to structure and guide the co-creation process. Living Lab participants were comprised of students, health promotion practitioners, researchers, and film and gaming experts who collaboratively designed and created the content and structure of the VR FestLab. Semi-structured interviews were conducted with nine students who participated in the Living Lab and represented young end users. Interviews were tape-recorded, transcribed and thematically analysed. Students described that they had influence on their tasks. They felt included and expressed that the collaboration with and feedback from peers and other stakeholders increased their self-efficacy and empowered them to take ownership and generate new ideas. Participants voiced that they lacked information about the final production of VR FestLab. Co-creation guided by the Living Lab methodology produced added value in terms of empowerment and increased self-efficacy for the students involved. Future Living Labs should plan for communication with participants about further development and implementation processes following ideation and prototyping phase.

## 1. Introduction

Participatory approaches have become an integral part of health research. This applies to health promotion research, where the aim is to improve the life of those who are subjects of research by actively involving them in various stages of the design and production of health promoting services, programmes or products [1]. One of the most impactful avenues for improving the development and implementation of health promoting interventions is to mobilise both explicit and tacit knowledge from multiple stakeholders and enable collaborative knowledge generation [2,3,4]. Shared learning between academic and community knowledge is fundamental for participatory research, which necessitates the establishment of equitable research partnerships with a diverse group of stakeholders such as public health professionals, health activists and community representatives [2].

Blending multiple stakeholders’ knowledge is widely used in the private sector, where consumers or end users are regarded as a valuable resource in design and planning processes because their experiences can help the company to improve both economic and societal benefits [5]. While stakeholder involvement is common in commercial practice, the involvement of stakeholders in prevention research remains limited. For example, Buyucek et al. [6] identified that stakeholder involvement was limited in formative research, implementation, and evaluation stages of interventions targeting problem alcohol use. Within the health care domain, community-based participatory research has emerged as the approach to blend a variety of knowledge from multiple stakeholders and communities with knowledge from academics [2,4]. It builds on a longstanding movement to promote patient-centred care, which seeks to empower patients as consumers [7]. Empowerment in this context is conceptualized according to Tones and Tilford [8] as a state in which an individual possess a relatively high degree of actual power and a genuine potential for making choices, which is associated with a high level of realistically based self-esteem together with a repertoire of life skills. In addition, the openness and collaboration addressed in participatory approaches may increase self-efficacy by improving the individuals feeling of confidence in performing and contributing to the development of the desired product [4,9]. 

The engagement of a broad array of stakeholders in various stages of the design and production process, has different origins but shares important assumptions and operating principals [3]. First, all co-creation models place emphasis on the process of co-creating and not only on creating the product or service. Furthermore, co-creation processes are guided by a systems perspective, which recognises the interrelationship between different parts of a system rather than focusing on any one part. Co-creation models place research as a creative enterprise that has human experience at its core. Finally, co-creation addresses the quality of relationships within stakeholder constellations, applying facilitation techniques that consider power sharing and utilise conflict as a positive force [3,4], thereby overcoming power structures which may reinforce the status quo.

Implementing and realising participatory co-creation is challenging, and different methods and systematic approaches have been applied to ensure the early involvement of users in innovation processes [4,10]. Few methodologies detail how to incorporate community and user values with supporting scientific evidence to demonstrate the full value of the process [11]. One example is Rundle-Thiele et al. [12], who outlined unique insights gained from dog owners that were illuminated with Dietrich et al.’s [13] six-step co-design process. Insights gained from dog owners were different from previous expert designed approaches aiming to reduce dog and koala interactions. David et al. [14] delivered empirical evidence showcasing how dog owner designed programmes can be implemented in community to change dog abilities to achieve the intended outcomes. These studies demonstrate scientific evidence of the capacity for co-design to deliver the desired outcomes. However, the co-design approach is only a specific instance of co-creation and focussed on gaining insights from end program users, excluding a broader array of stakeholders that require consultation in the co-creation process. 

An approach that has been described as a useful framework for realising participatory co-creation processes that involve a wide array of stakeholders is The Living Lab framework. The Living Lab methodology structures the design process into meaningful steps and is “a design research methodology aimed at co-creating innovation through the involvement of aware users in a real-life setting” [10] (p. 139). Thus, co-creation is a core activity that takes place in a Living Lab with diverse stakeholders and end users [15,16]. 

To overcome the challenges for developers of innovative products, services and programmes in the field of health promotion and education more research is needed about the important factors contributing to a successful co-creation process. Some research suggests that how co-creation activities are conducted, which users are involved, and how their involvement is facilitated is important for the outcomes of the process [17]. Furthermore, Voorberg et al. [1] found that 52% of the studies included in their review did not mention outcomes or outputs of the co-creation processes and concluded that if specific outcomes were reported, the focus was on the effectiveness of the products or programmes. Although participatory research includes young end users in various stages of the process, the majority of studies do not assess the outcomes related to youth perspectives of engagement and their experience as non-expert participants [1,18]. Thus, there remains a gap in the literature to understand perceptions of young end users following participation in a co-creation process. 

To add to the knowledge base on the potential benefits of co-creation processes for non-expert participants, the aim of this study was to investigate how young people perceived their participation in a co-creation process that involved multiple stakeholders developing a gamified virtual reality (VR) simulation targeted at adolescent users. Specifically, the focus was on understanding young participants’ perceived influence on the co-creation process as well as their perceived benefits and challenges.

## 2. Methods

### 2.1. The Co-Creation Process 

The Living Lab method was used to guide the co-creation process for the development of the VR FestLab. One assumption in the Living Lab is that co-creation takes place in real-life settings between a variety of stakeholders with different cognitive and motivational backgrounds [19]. In addition, Living Labs allow for obtaining user feedback and insights while experimenting. Using Living Labs as a framework to guide the co-creation process ensures that both explicit and tacit knowledge can be incorporated into products and services [10,15,20].

For the co-creation of the VR FestLab, a development group was formed comprising two prevention practitioners, two prevention scientists, two social marketing scientists, one VR game designer, two VR game scientists, one film production expert and eleven students who represented young end users. Students were recruited from a Danish folk high school that is specialised in film production and game design. The Danish folk high school provided the venue for the entire co-creation process. All workshops, the filming and follow-up interviews took place at this location. Two researchers facilitated the Living Lab framework using a six-step approach adapted from the Living Lab Handbook [16] and consisting of: (1) Exploration of key concepts, (2) Concept design, (3) Prototype design, (4) Innovation design, (5) Testing the product, and (6) Evaluation of the process and the product. This paper reports the initial three stages that are critical in the development of a first prototype. Stages 4–6 are focussed on improving the first prototype, which are ongoing at the time of writing and are not described in this article.

#### 2.1.1. Exploration of Key Concepts

Prior to the *first workshop*, the development group revised the existing film script from the Australian Blurred Minds alcohol education program [21]. The development group familiarised themselves with underlying concepts (gamification) and technology (virtual reality) and the outcomes envisaged for the gamified VR simulation, which aimed to strengthening self-efficacy and change attitudes towards excessive drinking [22]. During the *first workshop*, the development group was briefed on the Living Lab methodology. Next, the researcher and VR tool developer from Blurred Minds attended the workshop and contributed with knowledge and lessons learned from the development, delivery and evaluation of the Australian VR House Party game [22]. This was followed by an exploration of the Australian VR game using smartphones and headsets. The development group reflected on their experiences and shared insights into peer pressure and party scenarios. Importantly, the group developed alteration suggestions to create a contextually and culturally appropriate party setting. The outcome of this initial exploration and sensitising stage was a co-created list of elements that should be maintained, changed or added for a Danish setting. 

#### 2.1.2. Concept Design

At the concept stage, the students were invited to co-create a film script for the gamified VR simulation during their normal teaching hours at the folk high school. Based on the exploration of key ideas in the first workshop the students continued to work on the characters and their storyline in small groups of 2–3 people. Through student dialog and discussion, each scene was modified, revised and incorporated into the film script. The film production and VR game design experts facilitated this part of the process. Game design students developed ideas for mini games to be incorporated into the party simulation, which was not initially planned and emerged during the concept design stage. In the *second workshop* the students demonstrated the film script using role-play, illustrated the storyline of the gamified VR simulation via flow-charts, and presented the mini game concepts. These demonstrations and visualisations facilitated discussion and knowledge sharing with the other members of the development group (researchers, prevention experts and practitioners) and demonstrated authenticity of the process. At the end of the workshop, a list of improvements and changes for the film scripts was made. The concept stage resulted in a film-manuscript including descriptions of the characters to be casted, a comprehensive storyboard and ideas for mini-games to be included in the prototype.

#### 2.1.3. Prototype Design

The prototype stage involved 35 boarding school students (aged 15–17 years) serving as actors for the film-scenes. The students from the development group were responsible for casting and directing these student actors. Two full days were allocated for the filming using a 360-degree camera (GoPro Fusion). Students from the film production class in collaboration with the film production expert and the game design researcher conducted the shooting of the scenes for the gamified VR simulation. Postproduction was optimised with support from a professional film editor. The finalised scenes were integrated into a game engine (Unity, version 2019.2.5f1) to develop the interactive branching narrative. To improve the user experience the approach to prototype design was iterative including frequent user testing during the design process [23]. Lyk et al. [24] describes details of the technical realisation of VR FestLab. The outcome of the prototype stage was a beta-version of the gamified VR simulation. At this stage, the development group agreed on the final name of the prototype—VR FestLab. This decision was based on student suggestions from a session where two prevention practitioners and a group of boarding school students, after brainstorming and voting came up with two names. The prototype is detailed next. 

### 2.2. The VR FestLab Prototype 

VR FestLab consists of a computer simulation depicting a three-dimensional environment that shows a typical party situation for young people. As a game participant, it is possible to “steer” one’s own party through intuitive and realistic choices that can be carried out. While playing VR FestLab, the user is confronted with several behavioural options, where peers encourage the user to choose to either drink alcohol or choose soft drinks. Moreover, the user can choose to dance, play or support intoxicated peers (see Figure 1).

The behavioural options offered provide an opportunity for game players to make different decisions in a simulated, real-life setting. The options are designed to demonstrate the outcomes that occur with varying responses to peer pressure. Throughout the VR FestLab, the user can experience peers with positive and responsible behaviour regarding alcohol consumption. For example, getting a kiss, having a conversation with the older bartender and receiving positive peer feedback. On the other hand, the simulations allows users to explore the social costs of excessive alcohol consumption through a variety of consequences such as not being able to flirt with peers, not being able to go to the party at all or being confronted with negative peer feedback via texts received on the following day. 

### 2.3. Participants 

The current study used a qualitative approach with a purposive sample comprising of eleven students from the Danish folk high school who participated in the development group of the VR FestLab. Two students declined the invitation to be interviewed without further explanation and individual interviews were conducted with nine students from the development group (n = 4 females, average age 22.7, SD ± 3.5). Each student received a reimbursement of DKK 100 (EUR 13) for his or her participation in the interview. 

### 2.4. Data Collection and Analysis

Semi-structured face-to-face interviews with open-ended questions were used to generate data for the current study. Themes for the interviews focused on students’ opinion and feelings about participating in the two workshops (stage 1 and 2 in the Living Lab), their work and involvement before, between and after the workshops (stage 3 in the Living Lab) and their satisfaction with the collaboration. The main questions in the interview guide were: What was your role in the process?; How did you experience the process?; How did you find the cooperation to be, did you have a feeling of being an “expert” in this area and were your ideas being heard? Did you gain any experiences or learn something which you could use elsewhere?

A female MScPH student and research intern conducted interviews with nine students at the Danish folk high school in November 2018 shortly after filming of the scenes (stage 3 in the Living Lab). The interviewer participated in the two workshops in the role of an observer and note-taker and attended the filming in the same role, where she recruited the students for the interviews. The interviews were digitally recorded, and the length ranged from 25 to 45 m. 

Interviews were transcribed verbatim by another MScPH student-researcher, resulting in total 108 pages. Both data collection and the qualitative analysis were undertaken in Danish. To facilitate reporting, this research provides quotes translated from the original data. False names are used when referring to participants. Data were analysed focusing on the subjectivity, experiences and meanings as expressed by the respondents [25]. The software program NVivo 11 was used to facilitate the data analysis.

A researcher, who was not involved with conducting the interviews nor with participating in the co-creation process with students (steps 1–3 of the Living Lab), analysed and coded all the material using the four step process in systematic text condensation (see Malterud [26]). In the first step, all transcript pages were thoroughly read through to establish an overview of the data and identify preliminary themes. Thereafter, the data were organized into the meaning units and the codes were formed such as collaboration, attitudes towards the process and students’ specific contributions to the project. The coding was not based on the themes of the interview guide. Thirdly, the meaning units were extracted from the codes, and a few subgroups of codes were established. In the fourth and final step, the content of the data was reconceptualised [26]. All meaning units and codes were reviewed and discussed with a post-doctoral researcher from the development group, but not with interview partners.

The data were collected by a master student for use in a master thesis. No ethical approval was obtained for this data collection, because master theses do not require ethical approval in Denmark and at the University of Southern Denmark. The ethical standards of the Helsinki declaration were followed and written informed consent was obtained from all interview partners.

## 3. Results

The analysis of the interview data based on the first three steps in the Living Lab, resulted in four overarching themes, namely *general perception*, *influence, benefits* and *critical aspects* regards the students´ involvement in the co-creation process. Each theme is detailed in turn. 

### 3.1. General Perception of the Co-Creation Process

Students in general expressed that they were satisfied with the first workshop (Living Lab stage 1) and the visit by the researcher and VR game developer from the Australian project, from which the present project was inspired. This workshop facilitated the sensitising regarding the topic of alcohol prevention and allowed for trust to be built between all participants in the development group. Furthermore, students perceived the feedback from teachers, researchers and practitioners as timely, constructive and useful.

Some students expressed that their motivation to join the co-creation process was based on the novelty of the VR technology. However, some students did not recognise that it was voluntary to participate and perceived it as a natural mandatory task as part of their course work:


*When we were told what the project was about we were kind of forced into it, but I thought it sounded really exciting and interesting, because it is something which has never been done before. At least not here in Denmark. So mega interesting to be able to participate in the development of something completely new.*


### 3.2. Students’ Contributions and Influence on the Design of VR FestLab

Students felt to a high degree included and heard throughout the process, which made them even more motivated to participate in the co-creation process. Furthermore, it was a common experience among all interviewees that everybody contributed actively in the process. Emily explained how she felt included in developing the film characters:


*Our teacher told us about the different characters and then we could chose, depending on what we thought we would be good at, to work with what we wanted. I for example like writing manuscripts, so I chose to do that and to develop our characters.*


The students distributed the different tasks among themselves and each student had a high level of influence on their specific task regarding the storyline, manuscript, production and/or mini-games. 


*Even the silly ideas I came up with, they were also well received…[…]…As long as you kept within the boundaries of what was realistic, there were no wrong answers or wrong ideas. You could only come up with good ideas. I actually think that was really nice.*


Furthermore, the students were positively surprised that researchers and practitioners accepted the majority of their ideas and actually included them in the list of key concepts and also in the final storyline and film script.

In the beginning, students from the game development class felt insecure about their contribution to the project, since they regarded the gamified VR simulation more as an interactive film and not as game as such. However, within the process the students generated the idea of including mini-games, which gave the game development students a specific and tangible purpose. 


*There was not so much to do for game design, but this, the feeling decreased when we got the idea with the mini-games, because then we also had something more concrete to do as gamers.*


### 3.3. Student Benefits from the Co-Creation Process

Students expressed that the hands-on experience with film production and game-design (including the preparations, e.g., group discussions, idea-generation and story-board/flowchart) gave them confidence in their own skills and a feeling of optimism about future education and work, as expressed by Eric:


*Since I would like to become a game designer, it has given me some experiences to do this VR-project or those mini-games. And—of course the collaboration, we have worked really well together and that is something you can always use.*


The collaboration with and feedback from peers, teachers, researchers and practitioners during the co-creation process also enabled courage and self-confidence in the students. In addition, the pride of creating a product that lived up to expectations and would be used for an ethical purpose they supported (alcohol prevention among Danish adolescents) increased self-efficacy among students.


*I thought it was really fun to create these characters and it has given me a new kind of self-confidence.*


Furthermore, some students experienced that they could utilize their newly acquired skills in other school situations.


*Especially co-operating about developing the roles has taught me how teamwork works. But also about how you can structure a meeting where you can come up with ideas and pitch them. I used that the other day in a completely different course.*


In the second workshop (Living Lab stage 2) the students presented the first version of the film script of VR FestLab visualising the structure of the game through a flow-chart and using role-play to illustrate the dialogs written for each character in the film script. Several students, such as Lisa, felt that the appreciation and constructive feedback based on this visualisation from the rest of the development group, to the students, were better than they had anticipated: 


*I had expected that there would be more modifications than there were. That we had to change many things and perhaps had even misunderstood something. But it was very well received, and they really backed up our ideas. So, I think that was amazing.*


This positive feedback from the development group to the students further increased the students’ confidence both in themselves and in the group. As Noah explained:


*So I think that was cool, that us ordinary students who are eighteen or nineteen years old, can come up with something, which a grown-up man can approve and think is really good. That makes you feel kind of, put up on a pedestal, kind of like “Oh I know that I’m good”.*


Taken together, the students’ hands-on experiences with designing the gamified VR simulation, collaboration with peers and the research team, and the positive feedback from the development group built confidence in their own skills and generated optimistic feelings about their future professional education and work. Overall, they expressed feelings of courage, empowerment, self-confidence, and increased self-efficacy.

### 3.4. Challenges, Suggestions and Recommendations

Some students said that they did not fully understand or remember what the overall aim of the project was. However, the students expressed that they highly appreciated that the framework and tasks set by the teachers and researchers had few restrictions. This allowed the students to have a large influence on the content and structure of the game. Furthermore, students experienced some challenges in relation to the definition of tasks for both film production and game development students. During the process, the students organised their own tasks and new ideas were developed and executed. As Jacob described:


*At the beginning, we were not even supposed to do those mini-games. That was something we came up with while we created the characters and small manuscripts. That it [the mini-games] could be fun.*


While some students felt insecure about the process and the fact that they needed to define their own tasks, others perceived it as a fun challenge. Furthermore, some students, such as Sarah, suggested that the tasks should be more tangible and more explicitly divided between film production and game development students: 


*…The division of the tasks was done in a way where the film- and gamer-students were put in the same box. And we were given a common task instead of kind of dividing it between us…[…]…Perhaps [what we needed were] something more concrete which the gamer-students could do and something more concrete for the film-students.*


One student mentioned that the personal experiences and knowledge of students could have been better incorporated and used at the first workshop. 


*Perhaps at the first workshop, we could have talked more about our own experiences and then focus a bit on that.*


One student recommended informing students about the subsequent process of production and implementation of the gamified VR simulation. In addition, one of the students, Lisa, suggested that information on skills and competencies that may be achieved by participating should be provided to the students beforehand:


*In high school, before the exams you would read what you were supposed to know when you had finished a course or a class. To get the good grades you would have to fulfil those requirements. So in that way, you could [for the VR project] set up, not demands, but some ideas about what you can do when you have done this project…[…]….to set up some academic goals before the project had started…[…]….but then perhaps you would have too much focus on the academic goals and then forget about all the other things which you also learn.*


The quote above illustrates that the Living Lab presented an unfamiliar situation for students, who mostly have entered the course at the folk high school directly after high school. This cohort were not accustomed to such a participatory approach where predeterminations were not made thereby ensuring co-creation could take place. 

To sum, the most prominent challenges the students experienced while participating in the co-creation process, was a lack clarity regarding the overall purpose of the design task at hand, challenges in relation to the definition of tasks for both film production and game development students, and that students could have been more explicitly informed about the subsequent process of production and implementation of the end product (VR FestLab).

## 4. Discussion

While some studies have focussed on outcomes achieved from implementation of co-creation processes [12,14], less is known about co-creation participant experiences in prevention science. This study contributes to understanding that Living Lab participants receive benefits from participation including empowerment during the process and perceptions of enhanced employability. 

Using a participatory approach guided by the Living Lab methodology [10], students, practitioners, experts and researchers co-created the VR FestLab prototype. Thematic analysis of individual interviews revealed that students participating in the co-creation process felt empowered to engage in a dialog with practitioners and researchers about relevant and applicable content for the gamified VR simulation. Furthermore, the nature of the students’ involvement in the design of VR FestLab ensured a high level of influence on the design and production tasks, which increased students’ self-efficacy. This corresponds with several studies that argued that participatory research can empower participants to influence and generate novel ideas that will improve the creation of health products and services as well as their implementation in practice [4,17,27]. Empowerment became evident in this Living Lab process, where game design students took initiative and delivered the new mini-game components further enhancing the VR FestLab product. 

Participatory research often strives to increase the likelihood of developing more effective and efficient interventions that more precisely address health and well-being issues among the target group [27]. Studies have shown improvements in products developed in a co-creation process, especially on implementation and usability of the product or service [1]. However, the benefits for participants and other stakeholders may depend on the level of participation and how power and responsibilities for decisions are shared throughout the process. 

Studies have explored how to provide opportunities for involving children and adolescents in health promotion research [27]. One way to assess genuine participation by the children and adolescents in the development of health promotion interventions is using Shier’s model ‘Pathways to Participation’ [28], which describes five levels in relation to young peoples’ participation, namely that they are listened to (1), they are supported in expressing their ideas (2), their views are taken into account (3), they are involved in decision-making responsibilities (4), and they share power and responsibility for decision-making (5). Findings from this study’s interviews indicate a high level of participation of students (level 4 and level 5). Students experienced openness and opportunity from researchers, experts and practitioners to share power and responsibility for the decisions and choices related to the film characters and the development of the mini-games as new features of the end product. In addition, the joint problem solving among all stakeholders in the development group motivated the students to engage in the co-creation process [19]. From the student statements, it is evident that the students expected power asymmetry in the beginning. However, during the process they experienced more symmetry in power and decision-making as a result of their suggestions being acknowledged and accepted by the wider team. 

Some students expressed initial frustration, which later turned into confidence and mutual trust gained from the activities and the shared decision-making process. This progress in the collaboration and development of the tasks reflects that the co-design process depends on active involvement and ongoing facilitation [17]. In general, the students expressed satisfaction with the co-creation process and felt that they had a genuine influence in the decision-making process and design of the end product. However, the final technical production of the VR FestLab was the responsibility of the researchers in the development group that had the relevant technical skills. At this stage in the Living Lab, the active involvement by students in the process came to an end due to the fact that their course ended at the school. Thus, the students could no longer be involved in the further process of testing the first prototype and making suggestions for improvements or be involved in the final evaluation phase of the Living Lab process. Some of the interviews revealed a lack of information on the subsequent process of production of the VR FestLab and the benefits that their collaboration and work had provided in delivery of an end product. Planning and structuring a feedback loop that will provide this information should be incorporated in future Living Labs. However, this study shows that the framework of Living Lab was useful in order to structure the co-creation process into meaningful phases and to ensure that all stakeholders were invested and involved in the design and creation of the VR FestLab. Future research is needed to investigate the benefits of the Living Lab methodology on the effectiveness of co-created health promotion tools.

As a limitation, the study sample in this research only comprised of nine students who actively participated in the development group and therefore left out the researchers’ and practitioners’ perspectives. Data saturation as a criterion for sample size was not of concern for this study, because the base population is limited to eleven persons. The nine persons interviewed provided rich, appropriate and diverse data relevant to the research question, Hence, the generalizability of the research findings is limited to a specific study setting and a narrowly defined sample where some bias may be present given that two participants chose not to participate without giving their reason. Future research in a different setting is needed to further validate the presented findings and examination of all stakeholder participants is recommended. 

The students included in the development group were mainly aged 19–22. Therefore, they were on average four years older than the target group for the VR FestLab, which are adolescents aged 15–18. This may have introduced flaws to the content of the game due to retrospective recall bias related to their own drinking experiences. However, the actors casted for the filming were within the target group and this group revised language and actions during the shooting of the film scenes, thereby limiting potential biases for the endproduct. However, due to involving young adults rather than adolescents, our study does not fully represent the group of adolescent end users. Co-creation processes involving adolescents 15–18 years may experience more or other challenges than presented in our study and this offers an additional opportunity for future research. 

## 5. Conclusions

This study adds to the evidence on benefits for participants involved in the co-creation process. In line with previous studies [15,20], we found that using the Living Lab methodology increased the participants’ engagement, empowerment and self-efficacy. In addition, the Living Lab methodology enabled the co-creation and research to be conducted in an efficient manner without neglecting the involvement of and contributions from stakeholders. The results indicate that the co-creation process provided participating students with feelings of being capable and optimistic about their future capacities (empowered), interested and ready to collaborate (engaged), and listened to and accepted (understood). The importance of ensuring a feedback loop to communicate project outcomes with all Living Lab participants is indicated. 

## Figures and Tables

**Figure 1 ijerph-17-01097-f001:**
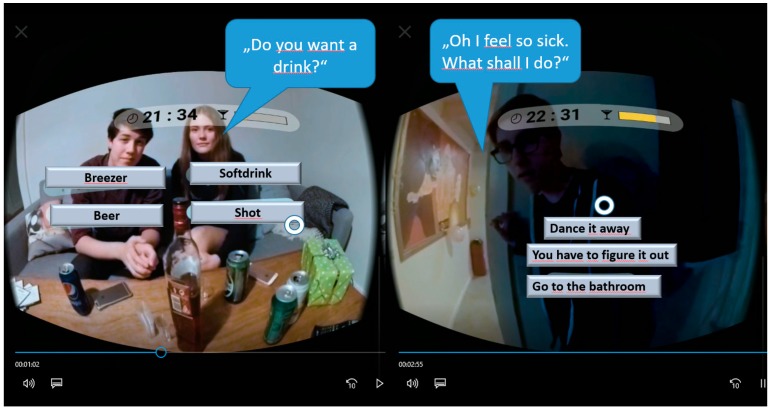
Screenshots illustrating two different decision points in VR FestLab.

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
