# Peer review of "Co-Creating a Virtual Alcohol Prevention Simulation with Young People"

_ijerph, 2020, doi:10.3390/ijerph17031097_

Round 1

Reviewer 1 Report

A brief summary (one short paragraph) outlining the aim of the paper and its main contributions.

This paper aimed to fill a gap in finding out how young people feel about participating in co-design and whether they benefitted from the process using qualitative interviews. The co-design process itself is also described in detail.

Broad comments highlighting areas of strength and weakness. These comments should be specific enough for authors to be able to respond.

This paper was quite interesting and novel. It does appear to be filling a gap in the research. The description of the co-design process, which itself was novel, was thorough and of interest. However, I feel that there is not enough description provided for the qualitative interviews conducted and I am slightly concerned that it seems only one person analysed this data. I have provided further specific in the scientific methodology section.

Reviewing Notes:

Originality/Novelty: Is the question original and well defined? Do the results provide an advance in current knowledge?

Yes, as highlighted in the broad comments I think this is a strength of this paper. Very interesting, and hopefully will stimulate more people to use co-design and further investigate the effects it has on participants.

Significance of content : Are the results interpreted appropriately? Are they significant? Are all conclusions justified and supported by the results? Are hypotheses and speculations carefully identified as such?

This is a fairly new/exploratory paper using initial qualitative approaches to start investigating a research gap. I feel that most of the conclusions are justified and supported by the results. The have appropriately acknowledge the limitations of the study.

Quality of Presentation: Is the article written in an appropriate way? Are the data and analyses presented appropriately? Are the highest standards for presentation of the results used?

Presentation is good.

Scientific Soundness: is the study correctly designed and technically sound? Are the analyses performed with the highest technical standards? Are the data robust enough to draw the conclusions? Are the methods, tools, software, and reagents described with sufficient details to allow another researcher to reproduce the results?

This is my area of greatest concern. Only having one person doing the analysis of the interviews I feel reduces the robustness of the results. There are many additional details around the qualitative approach that could be provided as outlined below using a reporting checklist. If these could be addressed I think that the paper would be robust enough and with the extra detail could be reproduced.

I used the COREQ (COnsolidated criteria for REporting Qualitative research) Checklist (http://cdn.elsevier.com/promis_misc/ISSM_COREQ_Checklist.pdf) during the process of this part of the review.

Personal characteristics: Item 1: It is not made clear which author performed the semi-structured interviews (refers to “a member of the research team” on line 205. Perhaps this could be added to the methods or author contributions at line 420. Therefore, and in addition, we also don’t know the credentials, occupation, gender, and experience of the researcher (points 2,3,4,5).

Relationship with participants: It should be the case that a relationship was established with participants through their participation in the co-design process. However, it is unclear if the researcher performing the interviews was involved in this part of the study. It is not mentioned if the participants knew the researcher, the reason for the interviews in particular. The characteristics of the interviewer could also be described (point 8).

Study Design: The rationale for the Living Lab method and the co-design was well explained for Point 9 and the systematic text condensation method was outlined. However, none of the results of the intermediate steps in this condensation process were reported in the results. Perhaps this was not felt necessary of useful.

Participant selection (Points 10,11,12,13) were generally well described. If available the reason for non-participation for the two participants who didn’t take part in the interviews could be provided but they might not have provided or recorded this information.

The setting is also described (points 14,15,16).

Data collection: A semi-structured interview was used with open-ended questions. I think it would be nice to provide the questions used somewhere (if there is more detail than the general themes available from lines 203-205. The use of prompting, guidance and pilot-testing (point 17) could potentially be addressed. Recordings were made (Point 19) and durations discussed (point 21). Not sure if any additional field notes were made or used (Point 20). Data saturation was not mentioned (Point 22).

Data Analysis: On line 214 it states that a single researcher analysed and coded the material (Point 24). I would feel more comfortable with the results of qualitative research if at least two people take part in this process. Or at least if there is a mention of a second researcher checking the themes developed, or a discussion of the themes amongst the research group. It seemed that themes were developed from the transcripts but perhaps some were in mind due to the themes in the interview and research questions (Point 26). Perhaps this could be made clearer. Software was mentioned (Point 27). It appears that the themes identified were not fed back to or checked with the participants (point 28).

Reporting: Nice quotes were provided for each theme (Point 29). Generally data and findings were consistent except for a few things that I have pointed out in the table of specific comments below which are mentioned in the Discussion without any evidence that I could find in the results (Line 412, 415). I feel the major themes were talked about in section 3.4 and some minor themes around challenges and suggestions in section 3.4 (Points 31 and 32).

Interest to the Readers: Are the conclusions interesting for the readership of the Journal? Will the paper attract a wide readership, or be of interest only to a limited number of people? (please see the Aims and Scope of the journal)

As stated above, hopefully this article could inspire more use of co-design as this is a methodology gathering more evidence all of the time. The use of VR environments for education in this setting is also novel and cool. Even if I didn’t have a specific interest in this paper I would click on it if I saw a summary on sciencedailynews or a similar website.

Overall Merit: Is there an overall benefit to publishing this work? Does the work provide an advance towards the current knowledge? Do the authors have addressed an important long-standing question with smart experiments?

I believe there is merit in publishing this work, with the additional of a few more details, It is more of a new question as more people move into using co-design, we need to know if it is beneficial to the participants as well.

English Level: Is the English language appropriate and understandable?

Yes.

Specific comments referring to line numbers, tables or figures. Reviewers need not comment on formatting issues that do not obscure the meaning of the paper, as these will be addressed by editors.

Just a few more specific comments below.

Line Number

Specific Comment

32

Future research directions are always outlined. You might consider stating what one good direction is or leaving this line out

339

I would like to see a summary of the actual results rather than a description of the aim/method. The sentence starting in the middle of line 361 for example could lead the discussion.

354

This section of the discussion is re-iterating the Introduction a bit too closely. Could potentially be removed or reworded.

384

The section in the discussion from 384 seems like results to me. The result was that they couldn’t find out what happened eventually. The discussion could be that this may be important feedback to provide in the future if they can’t take part in the whole process.

401

This paragraph introduces a large limitation, but nice to see it made explicit. Could perhaps be made clearer in the abstract?

412

Talking about the Living Lab methodology facilitating creativity. I don’t remember that being measured or mentioned elsewhere so may consider removing this, or providing more evidence earlier

415

Same for the time-efficient manner. This is the first I have heard of it, needs to be addressed earlier and evidence provided.

Reviewer 2 Report

First of all, praise the authors for the work done.The incorporation of gamification techniques to prevention provides a differential added value
On line 192, the authors refer to the participants, it would be interesting to include the age ranges, on which they worked
It would be interesting to include the limitations of the investigation. It is necessary to create occasions where young people can train skills
to resist social pressure, control their impulses or request help in relation to alcohol consumption, but it is also necessary to show the social desirability to which they are exposed in this type of research.

Round 2

Reviewer 1 Report

Thanks for your replies and revisions. They have addressed my previous comments very well and provided a clear response letter.

The only thing that I couldn't see in the revised manuscript (v2) was the revision mentioned below. There are no apparent revisions that I can see have been tracked in the results section starting in the middle of p. 6 through until the middle of page 8 apart from the addition of the word "creative" at line  330. This is the only place where the authors haven't provided the actual new text within the response letter so I wonder if this was missed.

Talking about the Living Lab methodology facilitating creativity. I don’t remember that being measured or mentioned elsewhere so may consider removing this, or providing more evidence earlier
Response: The facilitation of creativity is illustrated through the generation of new ideas (mini-games and film characters) which are mentioned on page 7 (line 262-263) and page 7 (line 272-275). To clarify we revised the wording in line 308-311 (page 7).

Thanks, and good luck with your future research!

Author Response

Talking about the Living Lab methodology facilitating creativity. I don’t remember that being measured or mentioned elsewhere so may consider removing this, or providing more evidence earlier.

Response: We have deleted the - in the first revision - inserted word "creative" in line 330. We have now deleted "facilitating creativity" from the conclusion part (page 7, line 438-439), as recommended:

"In line with previous studies [15,20] we found that using the Living Lab methodology increased the participants’ engagement, empowerment and self-efficacy."